# Effects of Rotavirus NSP4 on the Immune Response and Protection of Rotavirus-Norovirus Recombinant Subunit Vaccines in Different Immune Pathways

**DOI:** 10.3390/vaccines12091025

**Published:** 2024-09-08

**Authors:** Jingping Hu, Jinyuan Wu, Han Cao, Ning Luan, Kangyang Lin, Haihao Zhang, Dandan Gao, Zhentao Lei, Hongjun Li, Cunbao Liu

**Affiliations:** Institute of Medical Biology, Chinese Academy of Medical Sciences and Peking Union Medical College, Kunming 650118, China; hujingping@student.pumc.edu.cn (J.H.); wujinyuan@imbcams.com.cn (J.W.); caohan@imbcams.com.cn (H.C.); luanning@imbcams.com.cn (N.L.); s2023018021@student.pumc.edu.cn (Z.L.)

**Keywords:** rotavirus, nonstructural protein 4, recombinant subunit vaccine, immune method, norovirus P particle

## Abstract

Diarrheal disease continues to be a major cause of global morbidity and mortality among children under 5 years of age. To address the current issues associated with oral attenuated rotavirus vaccines, the study of parenteral rotavirus vaccines has promising prospects. In our previous study, we reported that rotavirus nonstructural protein 4 (NSP4) did not increase the IgG antibody titer of co-immune antigen but did have a protective effect against diarrhea via the intramuscular injection method. Here, we explored whether NSP4 can exert adjuvant effects on mucosal immune pathways. In this study, we immunized mice via muscle and nasal routes, gavaged them with the rotavirus Wa strain or the rotavirus SA11 strain, and then tested the protective effects of immune sera against both viruses. The results revealed that the serum-specific VP8* IgG antibody titers of the mice immunized via the nasal route were much lower than those of the mice immunized by intramuscular injection, and the specific IgA antibodies were almost undetectable in the bronchoalveolar lavage fluid (BALF). NSP4 did not increase the titer of specific VP8* antibodies in either immune pathway. Therefore, in the two vaccines (PP-NSP4-VP8* and PP-VP8*+NSP4) used in this study, NSP4 was unable to perform its potential adjuvant role through the mucosal immune pathway. Instead, NSP4 was used as a co-immunized antigen to stimulate the mice to produce specific binding antibodies that play a protective role against diarrhea.

## 1. Introduction

Rotaviruses (RVs), which belong to the family Reoviridae, consist of an 11-segment, double-stranded RNA genome lacking a lipid envelope. It is an important pathogen that causes diarrhea in humans, mammals, and birds [1]. RVs can be divided into 10 groups according to the difference in the VP6 protein of the virus [2,3] (A−J). Among all the groups, group A rotavirus (RVA) accounts for more than 95% of RVs infecting humans [4]. In addition, according to the difference in neutralizing antibodies against the outermost capsid VP7 and VP4 proteins [5], RVs are further classified into G and P serotypes. The main RV genotypes common in China are G1, 2, 3, 4, and 9, and the P genotype is dominated by P[8] [6]. Rotaviruses are still the main pathogens causing viral gastroenteritis in infants and young children [7]. Rotavirus gastroenteritis (RVGE) is highly prevalent in children < 5 years of age, and nearly every child has been infected with rotavirus at least once before the age of 3–5 years [8].

The rotavirus vaccines currently on the market include only live attenuated vaccines such as RotaTeq (Merck, NJ, USA), Rotarix (GlaxoSmithKline, Rixensart, Belgium), Rotavac (Bharat Biotech, Hyderaba, India) and LLR (Lanzhou Biologicals, Lanzhou, China), all of which are oral-route vaccines. These vaccines have a potential risk of causing intussusception in young children [9] and have been found to be more effective in high-income countries than in middle-income countries based on post-marketing vaccination data [10,11]. The precise reasons for the reduced efficacies are unclear but may be related to host mucosal factors and the gut microbiota [12,13]. For example, malnutrition and intestinal diseases may lead to a decline in immune function, which in turn reduces the ability of the body to produce antibodies after vaccination. Intestinal infection can also affect the immune response to live attenuated oral vaccines [14,15,16]. Therefore, the development of a vaccine that is administered via the parenteral route may make it possible to bypass the effects of the gut environment and achieve better protection [17]. In addition, non-replicating recombinant subunit vaccines do not contain genetic material and do not need to replicate in the body, making them safer than live attenuated vaccines.

During our previous experiment, we developed a subunit vaccine based on the norovirus P particle and rotavirus VP8* protein and nonstructural protein 4 (NSP4) [18]. VP8* is a subunit of the rotavirus outermost spike protein VP4, which is thought to be directly related to the binding of cellular receptors [19] (e.g., histo-blood group antigen receptor (HBGA) and sialic acid receptor) and has been verified to be a potent antigenic component [20,21,22]. NSP4 is a rotavirus nonstructural protein that plays a significant role in viral morphogenesis [5,23]. It plays important roles in viral replication, including (1) acting as an intracellular regulatory factor that transports double-layered particles (DLPs) into the ER to assemble the outer capsid proteins VP4 and VP7 [24]. (2) While acting as a viral pore protein channel that mediates the movement of calcium ions in and out of the cytoplasm [25,26,27], and as a viral enterotoxin that causes diarrhea, the C-terminus [28] of NSP4 is the main functional region, and the 114–135 aa or 112–175 aa peptides are thought to be the site of enterotoxin action [29,30,31]. (3) NSP4 has been found to have adjuvant properties [32,33], possibly recruiting macrophages by acting as a Toll-like receptor [34]. However, in our previous study, we reported that NSP4 had a limited effect on increasing VP8*-specific antibody titers after immunization via the intramuscular route and that antibodies against NSP4 did not have a neutralizing effect; however, binding antibodies against NSP4 alone could protect mice from diarrhea [18,35].

As a gastrointestinal virus, rotavirus primarily infects the human body through the intestinal mucosa; thus, we considered the mucosal immune route to confirm whether NSP4 acts as an adjuvant through the mucosal pathway. Mucosal immunization is an attractive strategy to prevent the transmission of influenza virus [36] and SARS-CoV2 [37,38] because it can induce systemic and local mucosal immune responses and stimulate the body to produce IgG and sIgA antibodies. Therefore, the effects of different immunization routes on vaccine efficacy were compared in this study. Four subunit vaccines, PP-V (with norovirus particles as vectors), PP-VP8* (with the VP8* protein inserted into the P particle), PP-NSP4-VP8* (co-expressing VP8* and NSP4), and PP-VP8*+NSP4 (mixed with NSP4), were designed on the basis of norovirus P particles. The P particle protein of norovirus is a subunit component of VP1 with three ring-shaped areas on the surface that are exposed to the outer surface of the particle, which can be used as an insertion platform for foreign proteins [39,40]. In addition, P particles can self-assemble to form the 24-valent subviral P24 nanoparticle, which can greatly increase the immunogenicity of exogenous antigens. The P24-VP8* recombinant vaccine was highly immunogenic in mice and gnotobiotic pigs [22,41]. Moreover, the P protein contains norovirus antigenic epitopes on its surface, which is an effective antigenic component of norovirus and is easy to express in large quantities in the *E. coli* system. Thus, P particles are good platforms for antigen presentation. In this study, NSP4 was inserted into or mixed with P particles to investigate the effects of NSP4 on VP8* and diarrhea in mice.

## 2. Materials and Methods

### 2.1. Antigen Preparation

The norovirus P domain was used as a platform for antigen presentation and the construction of three types of recombinant proteins. (1) PP-VP8* recombinant protein contains the VP8* protein of the rotavirus Wa strain (G1P8) and the P particles of norovirus (VA387). The VP8* protein was inserted into loop 2 (between T371 and D374) of the P domain of norovirus VA387. (2) PP-NSP4-VP8* recombinant protein contains VP8*, and NSP4 (NSP4_112–175_ from the rotavirus Wa strain) antigen with NSP4 inserted between the P particle and the VP8* protein. (3) P particles only constitute a control group. The DNA sequences of all three constructs were synthesized by Sangon Biotech Co., Ltd., and the pGEX-4T-1 vector (GE Healthcare Life Sciences) was used to clone these sequences. Another plasmid for the expression of the NSP4_112–175_ protein was also constructed and cloned into the pET-24b (Novagen) vector. The design of the antigens was based on published papers [18,35].

### 2.2. Recombinant Protein Expression and Purification

The *E. coli* (BL21, DE3) prokaryotic expression system was used to express four proteins (PP-VP8*, PP-NSP4-VP8*, PP-V, and NSP4) and induced with 0.4 mM isopropyl-β-D-thiogalactopyranoside (IPTG) at 16 °C overnight. PP-VP8*, PP-NSP4-VP8*, and PP-V were purified using Pierce™ Glutathione Agarose (Thermo Fisher Scientific, Waltham, MA, USA) via their GST tags. The AKTA protein purification system was used to purify His-NSP4 with a His-tag bound to Ni^2+^ resin (Cytiva, Marlborough, MA, USA), and a gel filtration chromatography system with a size exclusion column (Superdex 200, 10/300 GL, GE Healthcare Life Sciences, Pittsburgh, PA, USA) was used to remove the heteroproteins. Lastly, an ultrafiltration tube (3 kDa, Millipore, Burlington, MA, USA) was used to concentrate the protein.

### 2.3. Western Blot Analysis

Western blot analysis was used to identify purified recombinant antigens. After electrophoresis by 12% SDS-PAGE, the proteins were transferred to Immobilon-P transfer membranes (Millipore). The membranes were incubated in blocking buffer (10% skim milk in PBS) for 1 h. After being washed with PBST, the membranes were probed with rabbit anti-GST antibody (1:5000, Rockland Immunochemicals, Inc., Limerick, PA, USA) or mouse anti-His antibody (1:5000, Boster Biological Technology Co., Ltd., Wuhan, China) for 2 h at 37 °C or at 4 °C overnight. Then, the membranes were incubated with goat anti-rabbit IgG-HRP (1:5000, Multi Sciences Biotech, Hangzhou, China) for 1 h at 37 °C. After the addition of enhanced chemiluminescence (ECL) substrate (Multi Sciences Biotech), images were obtained using the corresponding software program Image Lab 5.2.1 from the Molecular ITmager ChemiDoc™ XRS + imaging system (Bio-Rad, Hercules, CA, USA).

### 2.4. Transmission Electron Microscopy (TEM)

The three recombinant proteins (PP-V, PP-VP8*, and PP-NSP4-VP8*) may automatically assemble into nanoparticles, and the morphology of the particles can be observed by TEM. The samples were absorbed to carbon-coated copper grids, using 2% phosphotungstic acid negative staining. The grids were then air-dried and inspected with a transmission electron microscope (Hitachi, Chiyoda City, Japan, H-7650) at an acceleration voltage of 80 kV.

### 2.5. Mouse Immunization

Rotavirus-specific antibody-free BALB/c female mice aged 6~8 weeks were raised in a specific pathogen-free (SPF) environment at the Small Animal Experiment Department of the Institute of Medical Biology, Chinese Academy of Medical Sciences (IMBCAMS). The mice were randomly divided into 11 groups of 6 mice each. Two immune pathways, intramuscular (IM) and intranasal (IN), were used, and the antigen dose was 10 µg/dose/mouse. The following immunogens were administered via the IM and IN routes: (1)/(6) PP-VP8* with phosphate-buffered saline (PBS); (2)/(7) PP-NSP4-VP8* with PBS; (3)/(8) PP-VP8* mixed with the NSP4 protein mixture; (4)/(9) PP-V protein as a platform control; (5)/(10) inactivated rotaviruses provided by the Li Hongjun Lab at IMBCAMS; and (11) phosphate-buffered saline (PBS, pH 7.4) was used as a vaccine diluent control. The mice were immunized intramuscularly or intranasally three times with 70 μL or 20 μL of immunogens without adjuvant at 2-week intervals. Two weeks after the final immunization, blood and bronchoalveolar lavage fluid (BALF) samples were collected, and the serum and BALF samples were processed using a standard protocol and then stored at −80 °C.

### 2.6. Detection of IgG and IgA Antibody Titers

For IgG antibody detection, VP8* or NSP4 was coated on 96-well plates (Corning, NY, USA) at 2 μg/mL with a flat bottom at a volume of 50 μL per well. The plates were incubated overnight at 4 °C. After washing with PBST, the plates were blocked with 5% (*w*/*v*) skim milk at 37 °C for 1 h, and incubated with 2 serial dilutions of mouse sera (diluted from 2000 to 4,096,000) at 37 °C for 1 h. Goat anti-mouse IgG-horseradish peroxidase (HRP) conjugate (1:10,000, Bio-Rad, Hercules, CA, USA) was used as a secondary antibody. However, for IgA antibody detection, the bound antibodies were detected with a goat-anti-mouse IgA–horseradish peroxidase (HRP) conjugate (1:10,000, Bio-Rad, Hercules, CA, USA). Lastly, the mixed substrate 3,3,5,5-tetramethylbenzidine (BD, San Jose, CA, USA) was added to the wells for 5 min, and after the addition of 2 mol/L sulfuric acid, the absorbance at 450 nm was detected with a spectrophotometer (BioTek Instruments, Inc., Winooski, VT, USA). The sera dilution with cutoff signals above OD450 = 0.15 were defined as IgG titers, and IgG titers with an OD450 lower than 0.15 at a dilution of 1:2000 were defined as 100 for calculations. IgA titers for which the OD450 was lower than 0.15 were defined as 1 for the calculations.

### 2.7. Fluorescence-Based Plaque Reduction Neutralization Assays

After treating the rotavirus (strain Wa) with trypsin (10 μg/mL) in Dulbecco’s Modified Eagle’s Medium (DMEM) for 20 min at 37 °C, the specific dilutions (dilution from 25 to 6400) sera from different immune mice were incubated with the rotavirus for 1 h at 37 °C. The sera and rotavirus mixture were added to MA104 cell monolayers in 96-well plates for one hour, after which the monolayers were washed with DMEM. Subsequently, the cells were incubated with DMEM lacking fetal bovine serum for 16 h at 37 °C in 5% CO_2_.

After being washed with PBS, the precooled 80% (*v*/*v*) acetone was added to the plates for 10 min at −20 °C. Subsequently, the plates were blocked with 2% (*w*/*v*) skim milk for 1 h at 37 °C. Following a further wash with PBS, our in-house-made goat anti-rotavirus antiserum (1:500) using as bound antibodies was added to the plates for a 1 h incubation at 37 °C. The plates were washed with PBS for three times, and then the fluorescein (FITC)-conjugated AffiniPure donkey anti-goat IgG (H+L) (1:200, Jackson ImmunoResearch, West Grove, PA, USA) was added to detected the bound antibodies for 1 h at 37 °C. After being washed with PBS, the plates were photographed with a Cytation 5 imaging reader, and a Gen 5 microplate reader and imager software 3.12 (BioTek, Inc., Crawfordsville, IN, USA) was used to counted rotavirus-infected cells in the form of fluorescence-forming plaques. The maximum dilution of sera at which fluorescence-formation plaques were reduced by 50% was defined as the 50% neutralization titers of mouse sera.

### 2.8. HBGA Binding and Blocking Assays

Type A saliva was collected from healthy people and boiled for 10 min. Following centrifugation, the saliva samples were diluted 1000-fold and used to coat a 96-well plate (Corning, NY, USA) at 4 °C. After washing the plates with 0.05% PBST (PBS containing 0.05% (*v*/*v*) polysorbate 20), 5% (*w*/*v*) skim milk was added to block the nonspecific binding sites at 37 °C for 1 h. After blocking, serially diluted PP-V (from 50 μg/mL to 0.05 μg/mL) protein was added, but the remaining well did not contain protein, and the plates were incubated at 37 °C for 1 h. A rabbit anti-GST antibody (1:30,000, Rockland Immunochemicals Inc., Limerick, PA, USA) was added to detect the bound particles. After washing with PBST, the HRP-conjugated goat anti-rabbit IgG (1:5000, Multi Sciences Biotech, Hangzhou, China) was added and incubated at 37 °C for 1 h. Then, the 3,3′,5,5′-tetramethylbenzidine (BD Bioscience) substrate was added, and the plates were incubated for 10 min at room temperature in the dark, and stopped by adding 2 mol/L sulfuric acid. A spectrophotometer (Bio-Tek Instruments Inc., Winooski, VT, USA) was used to detected the absorbance (450 nm).

PP-V protein at 5 μg/mL was preincubated 1:1 with serially diluted immunized mouse sera (from 25 to 6400, with no serum added to the last well) or BALFs (from 1 to 5, with no BALF added to the last well) at 37 °C for 1 h. The mixture was added to the 96-well plates coated with the boiled saliva and incubated at 37 °C for 1 h. The other processes were the same as those described above. The maximum serum dilution at which the absorbance at 450 nm was reduced by at least 50% compared to the unblocked positive control was defined as a blocking titer of 50%.

### 2.9. Suckling Mouse Rotavirus Challenge Model

To test the ability of the immune sera to protect suckling mice from diarrhea, 4-day-old BALB/c mice were used to measure their passive protection ability. In brief, human rotavirus (strain Wa) at 2 × 10^6^ plaque-forming units (PFUs) and rhesus rotaviruses (strain SA11) at 1 × 10^6^ plaque-forming units (PFUs) were preincubated with PBS (negative control) or various mouse sera (PP-NSP4-NSP4 sera were obtained from our previous experiment) at 1:15 dilution for 1 h before oral administration to suckling mice (*n* = 5 to 7). The diarrhea symptoms in mice were monitored daily for three days after the gavage by applying light pressure to the abdomen. Based on the scoring rule for suckling mice proposed by Boshutzenja et al. [42], the diarrhea of mice was scored from 0 to 4 points according to the color, hardness, and quantity of feces. A score of 0 indicates no feces discharge; a score of 1 indicates the formation of brown stool; a score of 2 indicates the formation of brown soft stool; a score of 3 indicates yellow soft stool; and a score of 4 indicates yellow watery stool or perianal fecal contamination. A score greater than 2 points was considered an indication of diarrhea.

Lastly, the ileum and jejunum were removed from 4-day-old mice and put into paraformaldehyde fixative.

### 2.10. Immunofluorescence (IF) to Detect the Distribution of Rotavirus in Intestinal Tissue

Fixed intestinal tissues were sent to Servicebio (Wuhan, China) for embedding into paraffin sections. The sections were deparaffinized with environmentally friendly deparaffinization liquid (Servicebio, Wuhan, China) and dehydrated using an alcohol gradient and distilled water. The samples were subsequently retrieved in 0.01% sodium citrate buffer (pH 6.0) (Servicebio, Wuhan, China) and permeabilized for 20 min with PBS containing 0.4% Triton X-100. The tissue sections were then blocked with 3% bovine serum albumin for 1 h at 37 °C. Following the blocking, the sections were incubated overnight at 4 °C with an in-house-made goat anti-rotavirus antiserum (1:500). The primary antibody was washed with PBS three times at 10 min, and then the secondary antibody fluorescein (FITC)-conjugated AffiniPure donkey anti-goat IgG (H+L) (1:200, Jackson ImmunoResearch) was added and the sections were incubated for 1 h at room temperature. After being washed in PBS, the slides were stained with DAPI (Servicebio, Wuhan, China). Lastly, the glass slides were mounted with Antifade Mounting Medium (Servicebio, Wuhan, China), and image acquisition was performed with a Cytation 1 imaging reader (Bio-Tek Instruments Inc.).

### 2.11. Statistical Analysis

GraphPad Prism 9.5.1 (GraphPad Software, La Jolla, CA, USA) was used to calculate the significance between several data groups by one-way analysis. A *p* value > 0.05 was considered nonsignificant (ns), a *p* value ≤ 0.05 was considered significant (marked as *), a *p* value ≤ 0.01 was considered highly significant (marked as **), and a *p* value ≤ 0.001 was considered extremely significant (marked as ***).

## 3. Results

### 3.1. Protein Expression and Purification

The fusion proteins of PP-VP8*, PP-NSP4-VP8*, PP-V, and NSP4 were expressed and purified using the *E. coli* expression system. SDS-PAGE (Figure 1A) and WB (Figure 1B) revealed that the sizes of the PP-VP8*, PP-NSP4-VP8*, and PP-V proteins were 65 kDa, 70 kDa, and 55 kDa, respectively. Their sizes were all as expected. The size of NSP4 was between 15 kDa and 20 kDa (Figure 1C). According to other research, we speculated that NSP4 may form multimers [43,44]. However, the concentration of NSP4 was very low, at nearly 1 μg/mL.

TEM was used to detect the morphology of the three recombinant PP proteins. The TEM results indicated that the PP-VP8*, PP-NSP4-VP8*, and PP-V fusion proteins self-assembled into nanoparticles (Figure 1D–F). Proteins self-assembled into nanoparticles may be easily discerned by immune cells and may improve the immunogenicity of subunit vaccines. In addition, foreign proteins that were inserted into P particles did not seem to destroy the ability to self-assemble. Therefore, these four proteins were used to evaluate their immunogenicity in mice and study the neutralization and passive protection of the mouse sera after vaccination with these vaccine candidates.

### 3.2. Results of Specific Antibody Titers from Serum and BALF Samples

#### 3.2.1. Detection of NSP4- and VP8*-Specific Serum IgG Titers

Sera and BALFs from different groups of mice were immunized with various immunogenic proteins by two immune pathways. We used indirect ELISA to determine the specific VP8* and NSP4 titers. According to the results shown in Figure 2A, the VP8*-specific antibody titer elicited by the PP-NSP4-VP8* nanoparticles was the highest among the groups, reaching 18,033, but there was no significant difference among the PP-VP8* (i.m.), PP-NSP4-VP8* (i.m.), and PP-VP8*+NSP4 (i.m.) groups. The results for VP8*-specific IgG are consistent with our previous finding that NSP4 cannot improve the IgG titer through the intramuscular injection method. However, the mice treated via nasal drip did not produce ideal serum-specific VP8* antibodies. The NSP4-specific IgG titers of PP-NSP4-VP8* (i.m.) and PP-VP8*+NSP4 (i.m.) reached 8000 and 10,683, respectively (Figure 2B), and there was no significant difference between the two groups. NSP4-specific antibodies could be detected in the serum of mice administered a nasal drip, but the level of these antibodies was low.

However, there was a significant difference (*p* ≤ 0.001) between the PP-VP8*+NSP4 (i.m.) and inactivated rotavirus (i.m.) groups, which indicates that inactivated rotavirus without an adjuvant cannot induce effective VP8* or NSP4 immune responses through intramuscular injection. Neither VP8*- nor NSP4-specific antibodies were detected in the PP-V and PBS groups. Therefore, the outcome of the intramuscular method is better than that of the intranasal immune method.

#### 3.2.2. Determination of NSP4- and VP8*-Specific IgA Titers in BALFs

The results of the indirect ELISA revealed that neither NSP4- nor VP8*-specific IgA titers were detected in BALFs except for those in the PP-NSP4-VP8* (i.n.) group. However, in this group, the titer was low. Compared with PP-VP8* and PP-VP8*+NSP4, PP-NSP4-VP8* may have a greater molecular weight and be more readily taken up by mucosal immune cells, thereby inducing IgA. However, IgA was detected only in the BALF of some mice, which was not statistically significant (Figure 3A,B).

The results of the antibody titers indicated that NSP4 does not exert an adjuvant effect through the muscle and mucosal immune pathways.

### 3.3. Results of Fluorescence-Based Plaque Reduction Neutralization Assays

Fluorescence-based plaque reduction assays were used to determine the 50% neutralization titer in mouse sera after vaccination with different vaccines. In this study, the immune sera from the three groups, PP-VP8* (i.m.), PP-NSP4-VP8* (i.m.), and PP-VP8*+NSP4 (i.m.), had neutralization effects on the Wa strain, at 1:25, 1:100, and 1:25, respectively (Figure 4A), and they were consistent with the results of the indirect ELISA which detected antibodies against IgG specific for VP8* (Figure 2A). These findings indicate that the greater the level of VP8* antibodies in the serum, the stronger the neutralizing effect on the rotavirus Wa strain. However, the results for the sera obtained by the nasal drip immune route were the same as those of the blank group, which were both negative. Therefore, the sera from the PP-VP8* (i.m.), PP-NSP4-VP8* (i.m.), and PP-VP8*+NSP4 (i.m.) groups should be used for the next step of in vivo neutralization experiments to determine their diarrhea-protective effects in mice. Four representative images of the major tested sera at dilutions of 1:25, 1:100, and 1:25 are shown in Figure 4B.

### 3.4. Results of HBGA Binding and Blocking Assays

HBGAs can be recognized by P particles, and they are important receptors for the norovirus infection of cells. Saliva samples from three healthy people with blood type A and different concentrations of the PP-V protein were used to select the optimal saliva sample and the optimal dilution gradient of the PP-V protein. The results are shown in Figure 5A. Saliva sample 2 was selected for the next experimental steps. The binding of the PP-V protein to HBGA receptors in saliva was optimal when the protein concentration was 5 μg/mL.

After the co-incubation of the sera and BALFs with the PP-V protein, the results of the binding saliva revealed that the sera from the mice that were intramuscularly immunized with PP-V, PP-VP8*, or PP-NSP4-VP8* all had 50% neutralization of P particles, with a serum dilution of 1:25 (Figure 5B). The dilution for the samples whose sera did not achieve 50% neutralization was set at 0. BALFs from intranasally immunized mice with PP-V, PP-VP8*, or PP-NSP4-VP8* showed no ability to neutralize P-particle protein. These findings indicate that the recombinant vaccines have the potential to protect against infection with both rotavirus and norovirus, which are two common gastrointestinal viruses. However, the protection afforded by serum is very limited and must be further enhanced.

### 3.5. Suckling Mouse Rotavirus Challenge and Immunofluorescence Results of the Small Intestinal Tissue Assay

#### 3.5.1. NSP4 Can Protect Suckling Mice against Diarrhea Caused by Rotavirus Wa Strain Challenge

The human-derived rotavirus Wa strain was less likely to cause diarrhea in mice, so we increased the gavage dose, and the suckling mice also developed different degrees of diarrhea, but it was generally slight, as shown in Figure 6C. The results (Figure 7A) collected on three consecutive days at 24 h, 48 h, and 72 h showed that the mouse sera following immunization with the PP-NSP4-VP8* antigen had better in vivo protection, and the mice did not develop diarrhea at 72 h, which was consistent with the results of the in vitro neutralization experiments (Figure 4A). In addition, the sera from the PP-VP8* and PP-VP8*+NSP4 groups had the same ability to neutralize the virus in vitro, and the protective effect against diarrhea in suckling mice was also similar, with 60% diarrhea at 72 h. The sera from the PP-NSP4-NSP4 group did not contain anti-VP8*-specific antibodies but also provided some protection against diarrhea, as only 60% of mice had diarrhea at 72 h. The diarrhea scoring is shown in Figure 7B. Most of the mice developed diarrhea score 3 or 2 as shown in the pictures in Figure 6C or Figure 6D, and very few developed watery stools, as shown the picture in Figure 6B.

The results of immunofluorescence can help us directly observe the quantity and distribution of RVs in the small intestinal tissues of suckling mice after a challenge. We dissected the suckling mice at 72 h after a challenge with rotavirus and collected small intestinal tissues fixed in paraformaldehyde, which were dehydrated, sectioned, and stained with DAPI and FITC, after which images of the small intestinal tissues were obtained via fluorescence microscopy. As shown in Figure 8, there were no significant differences in the viral load between the intestinal tissues of the mice given various sera and the virus mixture. However, compared with that in the positive control group, less fluorescence was detected in the intestinal tissues of suckling mice gavaged with a mixture of virus and sera from the PP-NSP4-VP8*.

#### 3.5.2. NSP4 Can Protect Suckling Mice against Diarrhea Caused by a Rotavirus SA11 Strain Challenge

The rhesus-derived rotavirus SA11 strain can cause severe diarrhea in suckling mice. The sera from different groups were incubated with rotavirus SA11 in vitro for 1 h and given to the mice by intragastric administration. The diarrhea from the suckling mice was observed for 3 consecutive days, and the diarrhea rate of the mice was calculated at 24 h, 48 h, and 72 h. The final statistical results revealed that (1) virus replication in the PBS group of the 4-day-old mice could lead to severe watery diarrhea. At 24 h, some mice (2/7) did not have diarrhea, and all the mice had watery diarrhea at 48 h, which was severe, as shown in Figure 6A. At 72 h, 71% of the mice (5/7) transitioned from having severe diarrhea (Figure 6A) to having mild diarrhea (Figure 6C). (2) The diarrhea of the mice in the PP-NSP4-VP8*, PP-VP8*+NSP4, and PP-NSP4-NSP4 groups consisted of mild watery stools (Figure 6C), a small amount of moderately watery stools (Figure 6B), and no severe watery stools. (3) Severe diarrhea of watery stools (Figure 6A) was observed in the PP-VP8* group at both 24 h and 48 h (16.7% and 33.3%, respectively), which shows that the in vivo protective effects of the PP-VP8* sera were not sufficient.

For the SA11 rotavirus challenge, mice in the PP-NSP4-VP8*, PP-VP8*+NSP4, and PP-NSP4-NSP4 groups had less severe diarrhea and no severe watery stools compared to those in the PP-VP8* group. The diarrhea status of each mouse was scored according to the BOSHUTZENJA scoring method, and the diarrhea mean scores are shown in Figure 9B. According to the statistical results, the probability of diarrhea after the gavage of the mice with the serum containing anti-NSP4 antibodies was lower, and the degree of diarrhea was less severe than that in the PBS group.

The IF results (Figure 10) revealed that when suckling mice were challenged with the rotavirus SA11 strain, in the absence of specific fluorescence in the negative control group (RV−), the positive control group (RV+) and PP-VP8* group presented obvious green fluorescence, whereas the green fluorescence of the PP-NSP4-VP8*, PP-VP8*+NSP4, and PP-NSP4-NSP4 groups were significantly lower than that of both the RV+ and PP-VP8* groups. NSP4, an important nonstructural protein in viral invasion, may act as an enterotoxin and an intracellular regulatory factor in vivo. Anti-NSP4-specific antibodies in the serum can bind to NSP4 and prevent rotavirus invasion in the body, thereby reducing the number of RVs in the small intestine.

## 4. Discussion

In recent years, with the introduction of the nasal spray influenza vaccine and the COVID-19 vaccine, we have considered the possibility of rotavirus subunit proteins working via the mucosal route. Since the mucosa is the largest immune organ of the body, the immunity caused by one mucosa can extend to the mucosal immune system throughout the body [45,46], including the gastrointestinal mucosa, the site of rotavirus invasion and replication. Therefore, mucosal vaccines are desirable. In our previously published study [18,35], we reported that NSP4 was not effective at increasing the titer of IgG specific for co-immune antigens (such as VP8*) when it was administered by intramuscular injection, suggesting that NSP4 could not play an adjuvant role in the intramuscular injection pathway. However, the ability to produce high levels of anti-NSP4 antibodies suggests that NSP4 functions as a co-immune antigen in the candidate vaccines. We therefore wanted to explore whether NSP4 could exert an adjuvant effect via the mucosal route (nasal drip).

In this study, we selected two proteins as target proteins, namely, VP8* and NSP4, and we used the P protein as the platform for inserting foreign proteins. VP8* proteins have been demonstrated to be potent target proteins for rotaviruses in several preclinical experiments and to contain multiple neutralizing epitopes. For example, the P24-VP8* vaccine confers strong protection against rotavirus diarrhea in gnotobiotic pigs [41,47], and the P2-VP8* vaccine is in phase II clinical trials [21]. Two vaccine candidates were designed in this study: one is PP-NSP4-VP8*, in which NSP4 and VP8* are tandemly linked and inserted into P particle proteins, and NSP4 is more likely to play an antigenic role as a macromolecular antigen. Another PP-VP8*+NSP4 vaccine candidate, in which NSP4 is a small-molecule protein, may be more inclined to exert adjuvant effects. Contrary to our expectations, the experimental results of this study (Figure 2A,B) revealed that the levels of specific VP8* and NSP4 IgG in the sera of the mice immunized via the nasal drop method were lower than those in the sera of the mice treated via the intramuscular injection route. In addition, only some of the sera from the mice in the nose-drip groups were able to detect specific VP8* and NSP4 IgG antibodies, and the number of samples was low, which may be partly due to differences in the operation. In addition, specific IgA was hardly detected in the alveolar lavage fluid of the mice (Figure 3). Thus, the three recombinant subunit protein vaccines without adjuvants used in this study could not exert an effective protective effect through the nasal drip route. Therefore, NSP4 does not act as an adjuvant through the mucosal immunization route.

In the present study, an antibody against NSP4 was confirmed to have a protective effect against diarrhea [30], probably because the binding antibodies against NSP4 blocked the enterotoxin effect of NSP4 and thus had a specific protective effect on mice in vivo [48]. The diarrhea statistics of the mice gavaged with the rotavirus SA11 strain (Figure 9) revealed that the probabilities of diarrhea in suckling mice in the groups gavaged with the mixture of serum containing anti-NSP4 antibodies (PP-NSP4-VP8*, PP-VP8*+NSP4, and PP-NSP4-NSP4) and the virus were lower than those in the PP-VP8*, PP-V, and PBS groups. Moreover, the degree of diarrhea was more inclined toward mild diarrhea, and severe diarrhea did not occur. The immunofluorescence results (Figure 10) revealed that the viral load in the intestinal tissues of the mice gavaged with the mixture of sera containing anti-NSP4 antibodies and the virus were lower than that in the PBS group or the PP-VP8* group, possibly because NSP4 is an important endotoxin for virus invasion.

Notably, while the PP-VP8* and PP-VP8*+NSP4 groups still presented 60% diarrhea after 72 h of Wa strain infection, diarrhea stopped in all mice (*n* = 5) in the PP-NSP4-VP8* group. The IF results (Figure 8) also revealed that fewer viruses were detected in the intestinal tissue of the PP-NSP4-VP8* group than in that of the other groups. Thus, the PP-NSP4-VP8* vaccine candidate could provide better protection against diarrhea caused by the rotavirus Wa strain. At the same time, PP-NSP4-VP8* demonstrated good immunogenicity with higher titers of specific IgG antibodies induced by PP-NSP4-VP8* in the absence of adjuvant compared to PP-VP8*+NSP4 and PP-VP8* (Figure 2), and only PP-NSP4-VP8* induced the production of specific IgA by nasal drip immunization (Figure 3). Furthermore, we paid special attention to the in vivo protective effect of the sera from suckling mice in the PP-NSP4-NSP4 group, which did not contain anti-VP8* antibodies. As shown in the diarrhea statistics (Figure 7A and Figure 9A), both for the Wa and SA11 strains, the sera in this group demonstrated some diarrhea protection in suckling mice, with 50% and 60% diarrhea levels at 72 h. These findings indicate that NSP4 can play a specific role as an antigen. Since NSP4 is a nonstructural protein with a more conserved sequence, the genetic similarity between the Wa and SA11 strains was 75%, whereas it was only 45% for the VP8* protein. The insertion of NSP4 into PP-VP8* may achieve simultaneous protection against rotavirus infection in multiple serotypes, making NSP4 a vaccine target site for continued development. In addition, antibodies against the norovirus P protein bind to HBGA receptors [49], thereby blocking norovirus from entering the cell. This advantage makes the recombinant PP-NSP4-VP8* subunit vaccine promising as a bivalent vaccine to prevent rotavirus and norovirus infections.

## 5. Conclusions

In conclusion, rotavirus NSP4 plays an antigen role instead of an adjuvant role in recombinant gastroenteritis virus-specific vaccines via both the intramuscular and intranasal immune routes. NSP4, which is a nonstructural protein, has a relatively conserved sequence, which provides protection against diarrhea caused by different strains of rotaviruses (the Wa strain and SA11 strain). In particular, the insertion of NSP4 into PP-VP8* provided optimal protection against diarrhea. However, the protective effect in vivo is not sufficient, and further enhancement of antibody titers is needed. And the mechanism by which NSP4 exerts protective effects in diarrhea also needs to be further studied.

## Figures and Tables

**Figure 1 vaccines-12-01025-f001:**
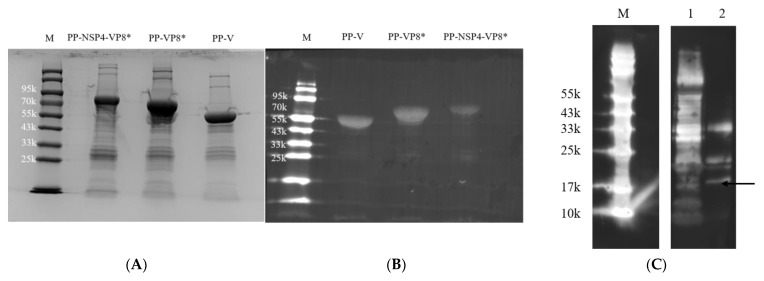
Production and characterization of the antigens. (**A**) SDS-PAGE result of PP-NSP4-VP8*, PP-VP8*, and PP-V; M represents the prestained protein markers with the indicated molecular sizes in kDa. (**B**) Western blot results for the three antigens. (**C**) Western blot results for NSP4; the black arrow represents the size of NSP4. Lane 1: Nickel column elution concentrate; lane 2: size exclusion column elution concentrate. (**D**–**F**) Representative EM micrographs of nanoparticles and aggregates assembled from PP-NSP4-VP8* (**D**), PP-VP8* (**E**), and PP-V (**F**). The typical subviral particles are indicated by black arrows.

**Figure 2 vaccines-12-01025-f002:**
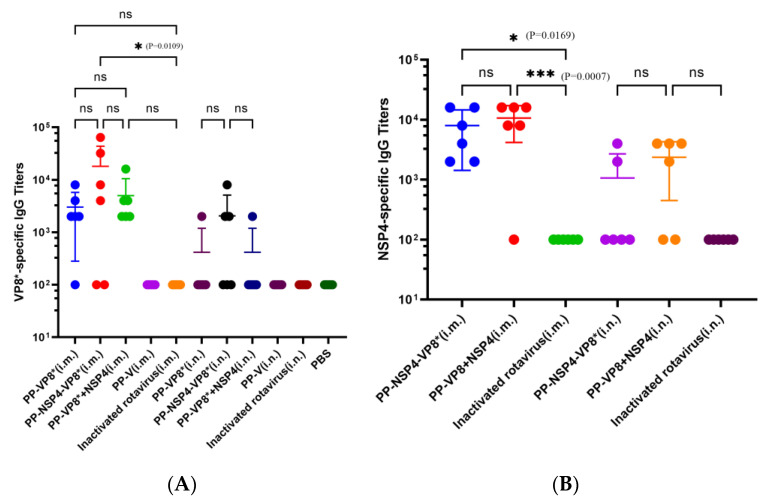
VP8*- and NSP4-specific IgG titers detected by enzyme-linked immunosorbent assay (ELISA). (**A**) Specific IgG responses of VP8*. (**B**) Specific IgG responses of NSP4. The differences between groups are indicated as follows: *, significant differences with *p* ≤ 0.05; ***, significant differences with *p* ≤ 0.001; ns, nonsignificant differences with *p* > 0.05.

**Figure 3 vaccines-12-01025-f003:**
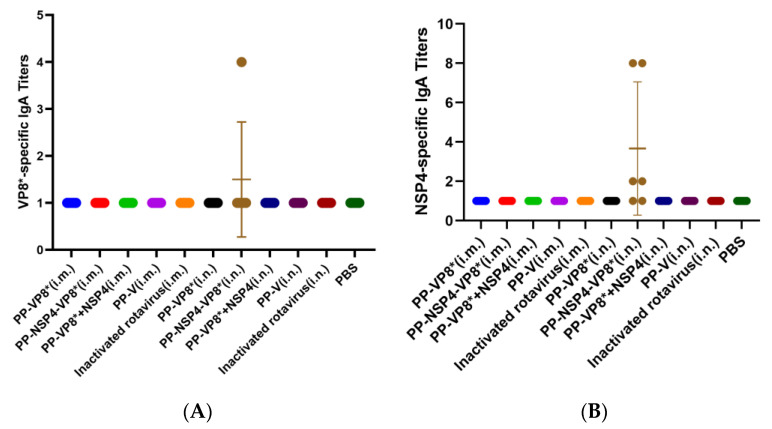
VP8*- and NSP4-specific IgA titers detected by enzyme-linked immunosorbent assay (ELISA). (**A**) VP8*-specific IgA responses. (**B**) NSP4-specific IgA responses.

**Figure 4 vaccines-12-01025-f004:**
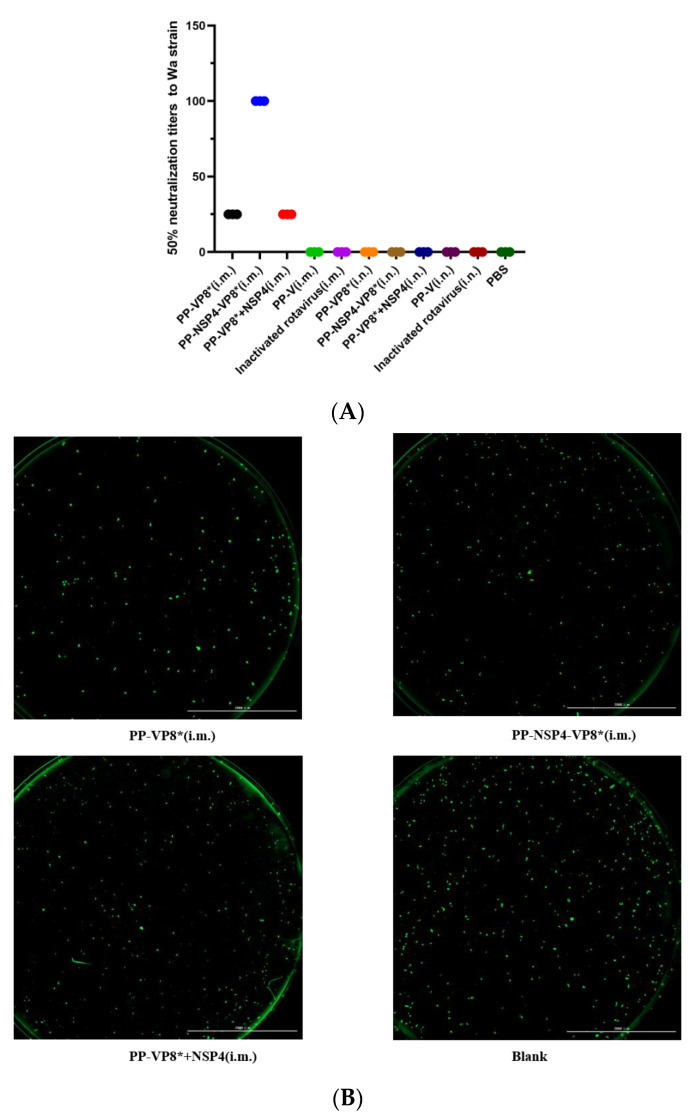
Neutralization of various vaccine-immunized mouse sera against the rotavirus Wa strain. (**A**) The 50% neutralization titers (*y*-axis) of mouse sera after administration with different immunogens (*x*-axis) against rotavirus (Wa strain) in cell culture were measured by fluorescence-based plaque reduction assays. (**B**) Representative images of the tested sera. The green spots represent the location of the rotavirus.

**Figure 5 vaccines-12-01025-f005:**
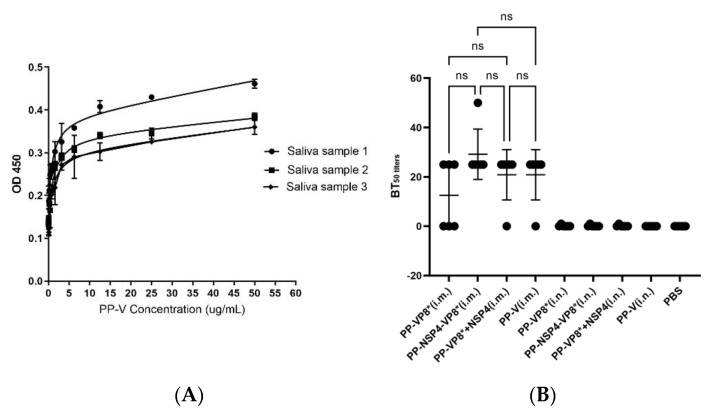
Binding curves and blocking effects of norovirus P particles in response to HBGA with various antigen−immunized sera and BALFs. (**A**) The binding curve of PP-V protein to type A saliva sample was used to select the appropriate saliva sample and PP-V subviral particle concentration for blocking experiments. The absorbance at 450 nm (OD450) reflects the binding ability. (**B**) Inhibitory effects on the 50% blocking titer (BT50) of various antigen−immunized mouse sera and BALFs. BT50 was tested with PP-V at 5 μg/mL and saliva sample 2. ns: non-significant.

**Figure 6 vaccines-12-01025-f006:**
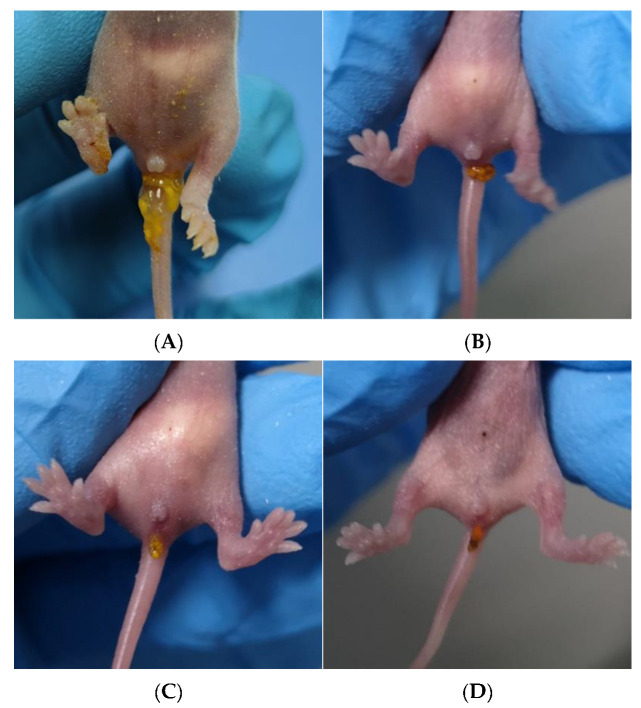
Four representative images of a suckling mice with diarrhea following a challenge with rotavirus. (**A**) Severely watery stool image, representing a score of 4. (**B**) Moderate watery stool image, representing a score of 3. (**C**) Mildly watery stool image, representing a score of 3. (**D**) Normal diarrhea image, representing a score of 2.

**Figure 7 vaccines-12-01025-f007:**
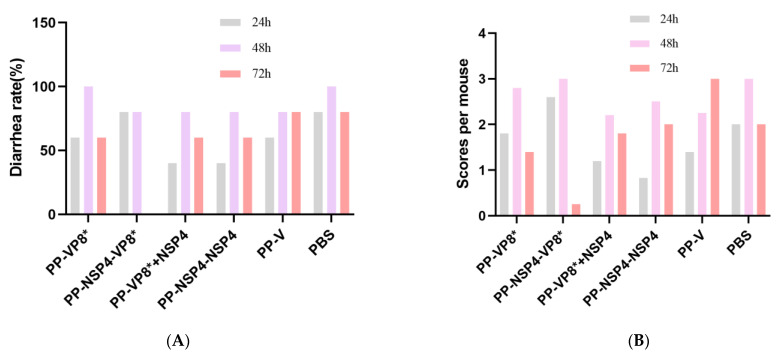
Passive protection of mouse sera immunized with various vaccine candidates against murine diarrhea caused by rotavirus Wa strain challenge. (**A**) Diarrhea rates (*y*-axis) of suckling mice after a challenge with the rotavirus Wa strain that was treated with mouse sera after immunization with various immunogens (*x*-axis). (**B**) Diarrhea scores of suckling mice after a rotavirus Wa strain challenge.

**Figure 8 vaccines-12-01025-f008:**
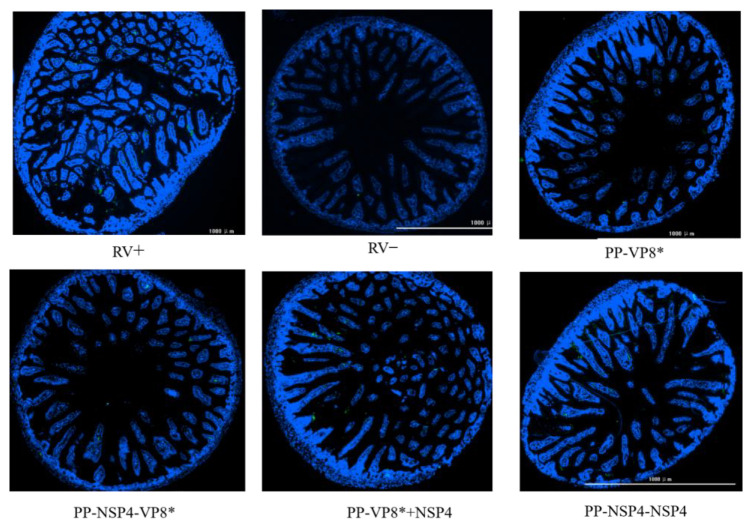
In vivo effects of VP8* and NSP4 on the rotavirus Wa strain. The immunofluorescence results revealed differences in virus loading in the intestinal tissues. The blue fluorescence represents the position of the cell nucleus, whereas the green fluorescence represents rotavirus. RV+: positive control group; RV−: negative control group.

**Figure 9 vaccines-12-01025-f009:**
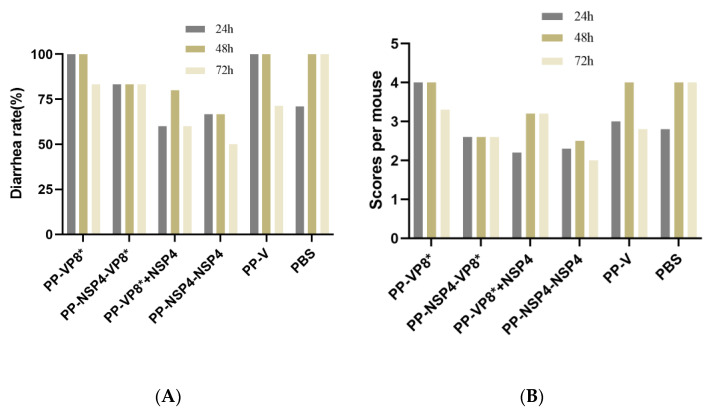
Passive protection of mouse sera immunized with various vaccine candidates against murine diarrhea caused by rotavirus SA11 strain challenge. (**A**) Diarrhea rates (*y*-axis) of suckling mice after being challenged with a rotavirus SA11 strain that was treated with mouse sera after immunization with various immunogens (*x*-axis). (**B**) Diarrhea scores of suckling mice after rotavirus SA11 strain challenge.

**Figure 10 vaccines-12-01025-f010:**
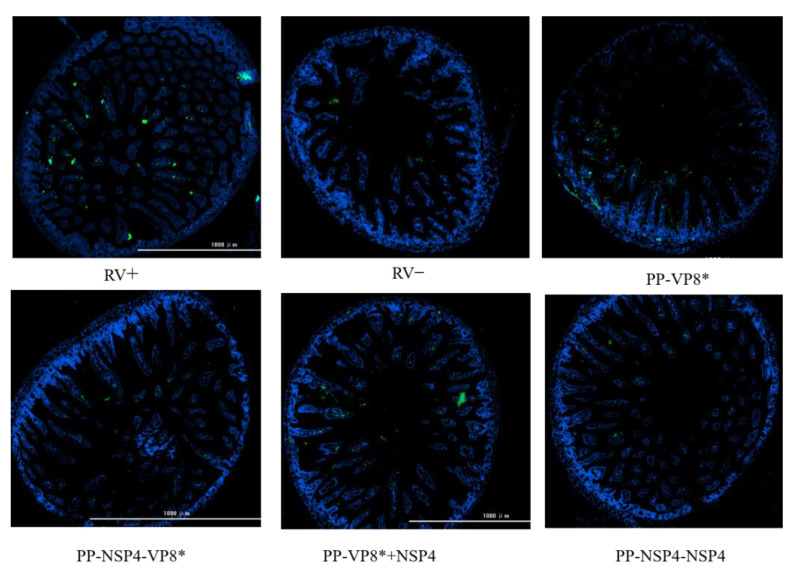
In vivo effects of VP8* and NSP4 on the rotavirus SA11 strain. The immunofluorescence results revealed differences in virus loading in the intestinal tissues. The blue fluorescence represents the position of the cell nucleus, whereas the green fluorescence represents rotavirus. RV+: positive control group; RV−: negative control group.

## Data Availability

All the data used during this study are available from the corresponding author upon request.

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
