# Peer review of "Effects of Rotavirus NSP4 on the Immune Response and Protection of Rotavirus-Norovirus Recombinant Subunit Vaccines in Different Immune Pathways"

_vaccines, 2024, doi:10.3390/vaccines12091025_

Round 1
Reviewer 1 Report
Comments and Suggestions for Authors
Generally, an interesting paper that deserves publication.
Comments on the Quality of English LanguageIn a few instances, the subject-verb agreement needs attention, such as "NSP4 were unable to perform its potential adjuvant role," where "were" should be "was" to match the singular subject. Some sentences are long and complex, breaking them into shorter, more concise sentences could improve readability.
Author Response
Comments 1: In a few instances, the subject-verb agreement needs attention, such as "NSP4 were unable to perform its potential adjuvant role," where "were" should be "was" to match the singular subject.
Response: Thanks for your suggestion,I have checked the grammar of the article again and corrected the mistakes.
Comments 2:Some sentences are long and complex, breaking them into shorter, more concise sentences could improve readability.
Response: Thank you for your suggestion, the article has been polished.
Reviewer 2 Report
Comments and Suggestions for Authors
The authors present an excellent study based on the construction of three types of recombinant rotavirus proteins for immunization using different routes, intramuscular and mucosal. The results include a high-quality data set of experments and among them, showed the advantages found in the recombinant PP-NSP4-VP8* subunit as a bivalent vaccine to prevent rotavirus and norovirus infections. However, these data were poorly addressed in the discussion. I suggest that the authors discuss these data in more depth. In addition, if possible, the authors should address data on neutralizing epitopes about all constructs presented in the manuscript from a structural perspective. The manuscript is suitable for publication. Below are some suggestions. Congratulations to the authors.
1. Introduction
Line 37 primarily infect mammals including humans
Fig 1 Lines
290-291: ....
(D to F) Representative EM micrographs of nanoparticles and aggregates assembled from 290
PP-NSP4-VP8*(D), PP-VP8* (E) and PP-V (F). The typical subviral particles are indicated 291
by black arrows..
- It's not clear what the arrows are highlighting...
Line 300:
the groups, reaching 18033....which scale
Author Response
Comment 1: Introduction Line 37 primarily infect mammals including humans.
Response:Thanks for your valuable suggestion,the incorrect statement here has been corrected.
Comment 2: It's not clear what the arrows are highlighting.
Response: The black arrows here represent nanoparticles formed by protein self-assembly, but the nanoparticles in the image are not particularly typical, possibly due to the protein concentration and the sharpness of the image.
Comment 3:Line 300: the groups, reaching 18033....which scale.
Response: The sera dilution with cutoff signals above OD450 = 0.15 were defined as IgG titers, 18033 represents the average dilution ratio of sera of six mice.
Comment 4: . The results include a high-quality data set of experments and among them, showed the advantages found in the recombinant PP-NSP4-VP8* subunit as a bivalent vaccine to prevent rotavirus and norovirus infections. However, these data were poorly addressed in the discussion. I suggest that the authors discuss these data in more depth.
Response: Thank you for your advice, i have added a description of the data related to the PP-NSP4-VP8* proteins in the discussion in line 670-676.
Comment 5: In addition, if possible, the authors should address data on neutralizing epitopes about all constructs presented in the manuscript from a structural perspective.
Response:I‘m sorry our lab has studied less about the structural aspects, so i have not explained the neutralization of proteins from the perspective of structure. I think this does not affect the integrity of this paper, and we will pay attention to increasing research in this area in the future.
We are thankful for your careful attention to detail and sincere opinions again!
Reviewer 3 Report
Comments and Suggestions for Authors
This paper reports an experimental mice study of the effects of the rotavirus nonstructural protein 4 (NSP4) on diarrhea, specifically whether the NSP4 protein can exert adjuvant effects on mucosal immune pathways. The study follows standard experimental protocols, the findings are interesting and useful, and the paper is well written.
The one suggestion that I have for revision is that to include in the Conclusions section a statement on the implications of the findings of the study for potential diarrheal control and on how additional follow-up studies should proceed.
Author Response
Comments 1: The one suggestion that I have for revision is that to include in the Conclusions section a statement on the implications of the findings of the study for potential diarrheal control and on how additional follow-up studies should proceed.
Response: Thanks for your suggestion, I have added relevant information in Line 696-700.